# A Review on Optoelectronical Properties of Non-Metal Oxide/Diamond-Based p-n Heterojunction

**DOI:** 10.3390/molecules28031334

**Published:** 2023-01-30

**Authors:** Xianhe Sang, Yongfu Wang, Qinglin Wang, Liangrui Zou, Shunhao Ge, Yu Yao, Xueting Wang, Jianchao Fan, Dandan Sang

**Affiliations:** 1Shandong Key Laboratory of Optical Communication Science and Technology, School of Physics Science and Information Technology, Liaocheng University, Liaocheng 252000, China; 2Ulsan Ship and Ocean College, Ludong University, Yantai 264000, China; 3Shandong Liaocheng Laixin Powder Materials Science and Technology Co., Ltd., Liaocheng 252000, China

**Keywords:** diamond, heterojunction, nonmetallic oxide, optoelectronic devices

## Abstract

Diamond holds promise for optoelectronic devices working in high-frequency, high-power and high-temperature environments, for example in some aspect of nuclear energetics industry processing and aerospace due to its wide bandgap (5.5 eV), ultimate thermal conductivity, high-pressure resistance, high radio frequency and high chemical stability. In the last several years, p-type B-doped diamond (BDD) has been fabricated to heterojunctions with all kinds of non-metal oxide (AlN, GaN, Si and carbon-based semiconductors) to form heterojunctions, which may be widely utilized in various optoelectronic device technology. This article discusses the application of diamond-based heterostructures and mainly writes about optoelectronic device fabrication, optoelectronic performance research, LEDs, photodetectors, and high-electron mobility transistor (HEMT) device applications based on diamond non-metal oxide (AlN, GaN, Si and carbon-based semiconductor) heterojunction. The discussion in this paper will provide a new scheme for the improvement of high-temperature diamond-based optoelectronics.

## 1. Introduction

Because of its excellent physical properties, diamond has become the final semiconductor material of the next-generation power devices, for instance, high carrier mobility (4500 cm^2^V^−1^ s^−1^ for electrons and 3800 cm^2^V^−1^ s^−1^ for holes) and high breakdown fields (>10 MV cm^−1^) [1]. Because it has a large band gap (5.5 eV), high-voltage resistance, extreme thermal conductivity, high radio frequency and high chemical stability, diamond has become a candidate material for optoelectronic devices working in high-power, high-temperature and high-frequency environments for case optoelectronic devices used in nuclear industrial processing and aerospace [2]. However, because the large-ionization energy of acceptor and donor impurities, it is still difficult to control the conductivity of n-type diamond, which hinders the emergence of new performance products. As a result, p-type diamond is usually used in photoelectronic devices for extremely harsh environments. Due to the close-packed structure and small internal voids induced by the short chemical bond of diamond, the traditional element doping technology usually causes serious lattice distortion of diamond and leads to deep-level doping, making it difficult to activate the carriers at normal atmospheric temperature. At present, the p-type doping technology of diamond is relatively mature, and the main dopant is boron atoms. Boron impurities can be easily integrated into natural diamond and microwave plasma chemical vapor deposition (MPCVD) diamond, and there is no crystal orientation problem. Diamond can be transformed from an insulator to a p-type semiconductor or even a superconductor by boron doping, where boron atoms exist as the main impurity in diamond [3]. For the past few years, various metal oxides (ZnO, WO_3_ and TiO_2_) and non-metal oxides (AlN [4], GaN [5], Si [6] and carbon-based semiconductors [7]) have been used in combination with p-type B-doped diamond (BDD) to form heterojunctions, which may be widely utilized in various optoelectronic device technology. For the devices of above, the formation of stable ohmic contact is an important aspect in the manufacture of diamond-based devices [8]. Diamond contacts should satisfy different requirements to be considered as good electrodes, such as a good mechanical adhesion, chemical and thermal stability over time. and the possibility of bonding connections [9]. Recently, the optoelectronic properties of diamond-based metal oxides were reviewed, and the research results of the past 10 years were discussed, which will promote the further development of high-temperature and high-power optoelectronic nano-devices. Up to now, all aspects dedicated to optoelectrical device applications of the non-metal oxide-based on diamond has not been explored. In this paper, we survey the past progress in device fabrication, optoelectronic performance research, LEDs, photodetectors, and HEMT of diamond non-metal oxide (AlN, GaN, Si, carbon-based semiconductor and other substrates) heterojunctions. Based on approximately 10 years of research, the existing problems in the research of diamond/nonmetallic oxide heterojunction optoelectronic devices are put forward, and the development prospect of this field is prospected, which will promote the birth of new high-temperature and high-power optoelectronic devices.

## 2. Diamond/AIN Heterojunction

The diamond/aluminum nitride (AlN) heterogeneous interface combines two kinds of large band gap materials with good performance, which is a research hotspot of optoelectronics, high-power, high-temperature and high-frequency microelectronics. Especially, AlN has typical characteristics of high breakdown field and high voltage, and its stable crystal structure is hexagonal wurtzite, which has the largest of nitride-forbidden band gaps ($E_{g}\;$~6.2 eV) [10,11]. The AlN/diamond heterojunctions have been applied to the first-principle calculations, high-frequency, field-effect transistors (FET) [12], high-power, light-emitting diodes (DUV LED) [13,14], and surface acoustic devices (SAW) [15].

### 2.1. Light-Emitting Diode

Khramtsov et al. put forward the idea of single photon-emitting diode (SPED) based on SiV center of nano-scale AlN/diamond heterojunction devices by various theoretical methods and demonstrated the data (Figure 1a). They found that despite the high-electron barrier in the AlN at the AlN/diamond heterojunction, electrons could be efficiently injected from the AlN into the i-type diamond region of the n-AlN/i-diamond/p-diamond heterostructure under the forward bias voltage (Figure 1b) [16]. Miskys et al. used plasma-induced molecular beam epitaxial growth AlN to achieve the AlN/diamond p-n heterojunction on (100) diamond, forming a heterobipolar diode with good rectification performance, which was efficiently luminous in the range from 2.7 to 4.8 eV spectra under forward bias conditions. This clearly confirms the feasibility of the heteroepitaxial growth of III-nitride on diamond and opens up interesting prospects for future optoelectronic devices working in far-UV light [13].

Hirama et al. studied the growth of the n-type single-crystal AlN (0001) layer on the diamond (111) substrate by metal-organic vapor phase epitaxy [17]. They observed that currents pour into emission at 235 nm wavelengths in an n-type AlN/p-type diamond heterojunction diode. The discharge is caused by free-exciton recombination in diamond. The AlN/diamond heterojunction has a valence band excursion of *E*_V_ of 4.0 eV and conduction band excursion of *E*_C_ of 3.5 eV, and the interlace type-II band permutation.

### 2.2. First-Principles Calculations

The alignment band at the AlN/diamond heterostructure was proposed for the first-principle study by Silvestri et al. They not only took into account the direction of AlN (0001) and diamond with (100) and (111) directions, but also speculated that the outermost AlN layer at the interface is composed of N atoms. The location of the interface carbon-nitrogen bonds is shown in Figure 2a, and it is clear that several misfit dislocations are made in each heterojunction. The experimental data display that the average excursion of the valence band is about 1.6 eV (Figure 2b), which matches the crisscross (type II) band alignment. It was also found that the valence band excursion has little dependence on the diamond orientation and strain.

According to the first-principles calculation on the basis of density functional theory, the early growth stage of AlN on the surface of oxygen-containing diamond (111) was discussed by Sznajder et al. They revealed how the strain existing in the system is released during the relaxation of the sedimentary layer. It is proved that due to the considerably large buckling of the top layer of the system, the blunt surface formed by the blunt surface of diamond (111) with various O and H atoms does not form a stable AlN structure [18]. Subsequently, they first imitated the uptake procedure of carbon atoms on the AlN {0001} exterior at the N- and Al- ends, focusing on inspecting the diamond/AlN exterior. Selecting some known spot flaws in the AlN and diamond structures, the height of the energy barrier met during the flaw transfer between the two materials at [0001] and (111) directions was calculated. These data indicate that the two components of the heterogeneous diamond structure/AlN may diffuse. The effect of spot flaws on the mechanic stabilization of the diamond/AlN interface was studied with the diamond/AlN heterostructure plate as a template [19]. In order to understand the engenderment mechanism of the hexagonal AlN on the exterior layer of cubic diamond (111), the structural characteristics of the single-crystal AlN (0001) layer at the early development stage were investigated. Afterwards, they put forward a model for the distribution of diamond (111) AlN (0001) atoms on the interface and expounded on the in-plane epitaxial relationship of the diamond (111) surface and the AlN (0001) layer according to the interfacial energy watched on the HR-TEM graphics (Figure 3) [20].

## 3. Diamond/GaN HEMTs

High-electron mobility transistors (HEMTs) made of gallium nitride (GaN) have been comprehensively researched since their birth. Contrasted with formal regular HEMT passage materials, for example silicon, GaN has a larger two-dimensional electron gas region (2DEG), a wider band gap, a higher high-frequency electron saturation velocity, and a higher breakdown voltage [21]. Diamonds are materials with a maximum heat conduction of 2000 W/mK, a high saturation rate and a transport rate for carriers and a large indirect band gap (5.5 eV). Therefore, GaN integration with efficient heat dissipation HEMT substrates can increase the thermal absorption capacity of GaN-based HEMT and reduce working temperatures, such as diamond.

In order to acquire high reliability and high device performance of GaN-based HEMTs, effective heat dissipation is very important, but it is still difficult. A great deal of sweat has been devoted to move a GaN equipment layer to the high thermal conductivity diamond substrate by bonding [22]. At room temperature, using silicon interlayers of 15 nm width and silicon interlayers of 22 nm width, two types of GaN diamond-bonded composites were formed by improved surface-activated bonding (SAB) (Figure 4). This high thermal stability proves that the GaN template on diamond can be formed by normal temperature bonding, which can be applied to equipment processing and further extension growth. Yaita et al. used the high thermal conductivity of a microcrystalline diamond (MCD) film to improve the heat dissipation of GaN HEMTs [23]. Diamond films can be grown at high temperatures from the thermally stable metal insulator semiconductor (MIS) grid structure of GaN HEMT’s. Therefore, the MCD thin membrane directly on the outer layer of GaN HEMTs not only improves thermal resistance of the HEMTs, but also does not cause grid structure loss.

Zhang et al. embedded a manifold microchannel cooling (EMMC) device to reduce the junction temperature, in which the direct etching of the microchannels on the GaN substrates was used to absorb the heat produced from itself. In order to effectively improve the spread of the heat efficiency of the junction, the covered EMMC GaN device and the diamond-heated spread device are connected together to form an integrated device. The newly type EMMC configuration, together with diamond capping, is expected to be the final near-junction cooling solution, which will help to realize and develop the high-power compact GaN-based devices [24]. Ranjan et al. first reports on the use of conductance methods to investigate the detection at distinctive temperatures (25–200 °C) of the hetero-contact surface trapping trait in AlGaN/GaN by a high HEMT on CVD diamond [25]. The main pitfalls at the AlGaN/GaN heterointerface are considered to be fast pitfalls with time constants in the interval length of 0.16 to 10.01 μs. The trap density (DT) was found to increase with the temperature because at high temperatures the deep pitfalls in the band gap are excited. Lastly, the DT behavior is related to the current collapse phenomenon in the device by the temperature-related pulsed *I_DS_–V_DS_* measurements (Figure 5).

The formation of HEMTs using AlGaN/GaN and diamond as raw materials and the SiNx interface layer grown in situ as two-terminal was studied by metal-organic chemical vapor deposition (Figure 6). The effective isotropic thermal conductivity of the diamond was $176_{-35}^{40}$ W/mK measured by the time-domain thermos-reflectance (TDTR). The efficacious isotropic thermal conductivity of the diamond/GaN interface (including the attached layers and SiNx) measured by the time-domain thermal-reflectance (TDTR) is$\;52.8_{-3.2}^{5.1}{\;m}^{2}$·K/GW. Using the finite element method, three-dimensional thermal simulation was constructed to analyze the heat dissipation of diamond with AlGaN/GaN HEMT as substrate. The temperature distribution and heat propagation path in the chip were studied. The results showed that the width of the extend layer and the thermal resistance of the contact interface have some effects on the junction temperature, especially the thermal resistance of the contact surface, whose impact is up to 19 K per 10 m^2^ K/GW, which is due to the level and perpendicular thermal diffusion capacity near the junction area (Figure 7).

The combination of diamond with the high breakdown strength of electric and thermal conductivity and GaN shows great prospect in finding effective thermal management solutions. In the past few years, both academia and industry have been trying to find a technological process that can integrate the two materials and manufacture high stability and excellent functioning new machines. Many methods have been suggested, for example, developing diamond films to cover passivated GaN equipment, or diamond/GaN wafers for new generation equipment research and development. The prospect and difficult problem of science and technology of each method are reviewed by Mendes et al., and their advantages and disadvantages are introduced and discussed in detail [26].

## 4. Diamond/Si Heterojunction

Due to the fact that the most low-priced electronic products and most electronic products on the market use silicon, silicon is widely applied in optoelectronic applications. The growth of a polycrystalline diamond layer on silicon substrate by vapor deposition is a great prospect material for project sensor equipment (Figure 8). It can properly adjust the deposition parameters properly to form the diamond layers with different a deficiency structure and crystallinity.

A new method has been proposed to describe the conductivity mechanism of the undoped CVD polycrystalline diamond layer on silicon substrate by Szymon et al. Through the *I-V-T* characteristics of this material in the current-voltage-temperature range from 77 to 500 K, the valid data of this material’s conductivity were obtained. *I-V-T* data were recorded in the positive configuration of the n-Si/p-CVD diamond heterojunction, which indicates that the jump groove defect is the main mechanism that limits the conductivity (Figure 9) [27].

Undoped polycrystalline diamond films (PDFs) were formed on n-type silicon (Si) using the hot filament chemical vapor deposition method (HFCVD). Based on the temperature-dependent current-voltage (*J-V/T*) features, the affection of the hydrogenation degree on the electrical specific of the p-diamond/n-Si heterojunction was studied. It was shown that the hydrogen effect reduces the energy distribution in the crystal interface and reduces the cut of the bore. It was shown that the hydrogen effect reduces the energy distribution in the crystal interface and reduces the cut of the bore. Hydrogenation could significantly reduce the GB potential barrier [28]. This n-type NCD/p-type Si heterojunction diode was fabricated by microwave (MW) plasma enhanced CVD, and its electrical transport characteristic were studied by Kungen et al. At the temperature up to 473 K, the rectification characteristics of the reverse bias area and forward bias area were investigated to deepen the understanding of the carrier transport principle according to the forecast band Chart. The disordered amorphous/Si heterostructure interface is mainly related to the grain boundaries in amorphous films (Figure 10) [29]. The high ideal coefficient and small current injection barrier was used to describe the current-voltage characteristics.

The n-type (nitrogen-doped) supercrystalline diamond (UNCD)/hydrogenated amorphous carbon (A-C: H) composite film was grown on a p-type Si substrate to prepare a heterojunction diode in a mixture of nitrogen and hydrogen. The results show that the conductivity is mainly controlled by GBs but not by UNCD grains. Moreover, the extracted value of the permittivity of the n-doped UNCD/a-C: H films is comparable to that of the microcrystalline diamond, indicating that the capacitance contribution in the device is mainly derived from the UNCD particles. This study demonstrates the ability of the impedance profiles to provide a clear separated contribution of the sp^3^-and sp^2^-bonded carbon to the conductivity of coexisting materials [30]. The features of the p+-Si/P-diamond heterojunction diodes (HDs) created using surface-activated bonds with those of the Al/P-diamond Schottky diodes (SBDs) made on the identical diamond substrate and the temperature dependence of the *I-V* and *I-V* peculiarity of the P+-Si/P-diamond HDs were investigated by Uehigashi et al. At the annealing temperature of 873 K, the ideal coefficient, on/off ratio of the HDs and reverse bias current all become larger, which is consistent with the peculiarity of the SBDs, and they form a reasonable contrast [31]. Annealing reduces the energy barrier height of the diamond/silicon bonding interface. SiV has been widely studied in various diamond color centers, and it is expressed as a different charge (SiV^0^) or a negative pressure (SiV^−1^). Its different zero suppression cables (ZPL) can distinguish these two charges. Up to now, it is still a great challenge to convert the neutral charge kind of diamond into the negative charge type. To solve this problem, (100) microcrystals containing silicon vacancy (SiV) centers were used to form diamond films on the n-type silicon substrates to form the diamond/n-Si heterojunctions. The diamond/n^+^-Si heterojunctions show that it about two orders of magnitude of the rectifying ratio. In the diamond/n+-Si heterostructures, the intensity of the SiV^-^ photoluminescence (PL) stands under reverse bias, while under forward bias, the intensity of the SiV^-^ PL increases twice, which is greater than that of the diamond/n-Si heterostructures (Figure 11) [6].

## 5. Diamond/Carbon-Based Semiconductor Heterojunction

With the size of silicon-based semiconductor devices approaching the physical limit, silicon flexibilization has become increasingly close to the flower board. The breakthrough of carbon-based materials provides a better choice for flexible electronics. Carbon-based sheet materials are hot candidates for the next generation of optoelectronics and electronic devices (including photodetectors, light modulators, and transistors) have attracted great attention. Among them, carbon nanotubes and graphene are recognized as the “natural choice” materials for flexible electronics due to their excellent electrical properties, light transmittance, especially ductility [32].

A graphene/n-type diamond heterostructures was prepared and it was hydrogen-terminated and diamond-doped with heavy phosphorus by a wet-transfer process. The site emission principle and energy band structure of the graphene/n-type diamond heterojunction was investigated via the site emission electronics energy spectrum and ultraviolet photoelectron spectroscopy (UPS). UPS data indicate that the upward energy band bending at the near point of the graphene-diamond interface forms an internal barrier in diamond. The electron emission limited by the surface terminal is transformed into graphene directional emission with the formation of the heterojunction, and the electron emission limited by the surface terminal is transformed into graphene directional emission (Figure 12a, b). The thermal and electrical characteristics of a typical diamond-on-graphene FET transistor made of nanosheets were studied by Bogdanowicz et al. It is found that at low temperatures, the thermal activation transport on the barrier was controlled by quantum tunneling (Figure 12c). These results proved the low-temperature diamond-on-graphene heterostructure devices were realized for the first time.

In nitrogen and hydrogen-mixed gases, nitrogen-doped ultrananocrystalline diamond/hydrogenated amorphous carbon (UNCD/a-C:H) thin membranes are developed using coaxial arc plasma deposition. Within the bias voltages range from −5 V to +5 V, *I-V* curves exhibited a correcting behavior well in the dark. At monochromatic light at 254 nm, the heterojunction shows UV-sensing ability, probably because of the presence of UNCD particles. The experiments show that the nitrogen-doped UNCD/a-C:H membrane can be used as photovoltaic materials in the ultraviolet range (Figure 13) [33]. A heterojunction composed of graphene and microcrystalline diamond (MCD) was fabricated by a diamond thin membrane and the whole wafer transfer process, demonstrating a UV detector. Graphene cannot be deposited on the top surface of the MCD thin membrane via the traditional direct transfer process. Nonetheless, it is indicated that the 2 μm-thick MCD diamond thin membrane used in the graphene transfer process of high quality graphene/MCD heterojunctions can be effortless stripped from the growing silicon substrate and has a flat back to be used as a platform for future new optoelectronic systems.

## 6. Conclusions and Future Outlook

In this paper, the latest research results of non-metal oxide/diamond-based pn heterojunctions are studied, particularly those that comprised significant types of non-metal oxide (AlN, GaN, Si and carbon-based semiconductor), including device fabrication, optoelectronic performance research, first-principle calculations, LEDs, photodetectors, and HEMT device applications. The reports mentioned above greatly expand the background understanding of the optoelectrical characteristics of non-metal oxide/diamond-based pn heterojunctions in optoelectronic devices. Moreover, research personnel are seeking new manufacturing routes and electrode types to raise the performance of this equipment, such as improving the preparation of non-metal oxide/diamond-based pn heterojunctions in order to raise the reaction speed of optoelectronic devices, combined with different device structures, which provide a promising scheme for optoelectronic applications. Here is a list of non-metal oxide diamond pn heterojunction-related efforts that have not been explored in depth (Table 1).

The biggest challenge for optoelectronic devices used in harsh environments is their life and stability. In the past long-term research, the properties of optoelectronic devices of non-metal oxide/diamond heterojunction have reached a high level, but the stability and life of devices decrease with the increase of temperature, and the failure rate of equipment increases with the increase of temperature. In addition, the current transmission mechanism is closely related to temperature. The carrier mobility, carrier generation rate, and intrinsic carrier density are related to temperature. The problem of power loss and reducing circuit behavior related to temperature can be solved by developing stable high-temperature nonmetallic oxide/diamond heterojunction optoelectronic devices, for example conductance and diode voltage drop. As a result, an attempt should be made to research high-temperature optoelectronic performance, which can increase the properties of photoelectric equipment, containing photodetectors, LEDs, photodetectors, and HEMT device applications for radiation-proof, high-power and high-temperature environments.

## Figures and Tables

**Figure 1 molecules-28-01334-f001:**
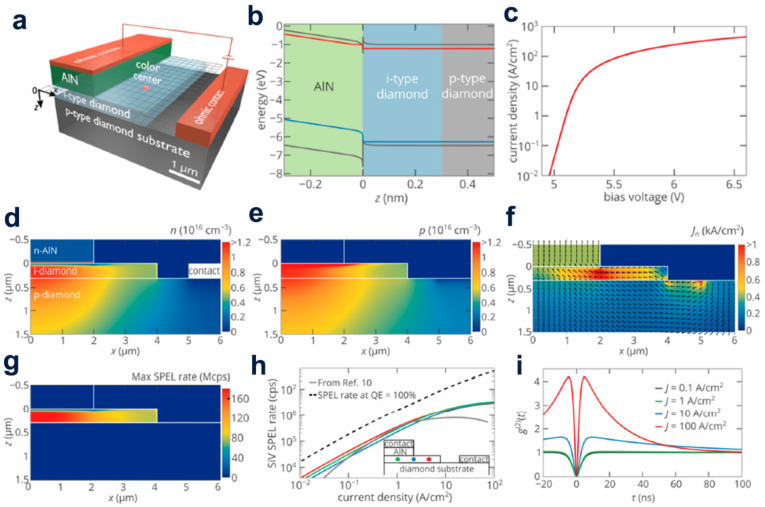
(**a**) Schematic diagram of AlN/diamond Single Photon Light-Emitting Diode (SPED). (**b**) Energy band diagram of near aluminum nitride/diamond heterojunction at 6.5 V. (**c**) The relationship between the current density of the top n-type contact and the bias voltage of the speed. The electron density (**d**), distribution of hole density (**e**) and electron current density (**f**) in speed when the bias voltage is 6.5 V. (**g**) Correlation between the single photon electroluminescence (SPEL) tempo of the SiV center, and displays its place and velocity in the i-type diamond area. (**h**) Relationship of the trait of the SPEL speed at the center of SiV on the injection current at three distinct locations in the core of the i-region as displayed in the insertion diagram. (**i**) $g^{\left(2\right)}$ for different pumping levels, at x = 1  μm, z = 150 nm.

**Figure 2 molecules-28-01334-f002:**
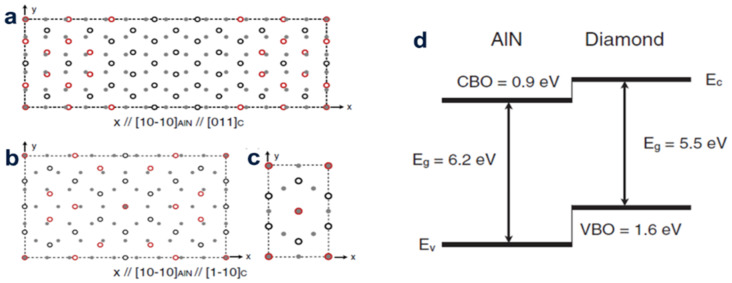
The vertical view of the supercell imitating the heterojunctions$\;H_{1}$ and $H_{2}$ (**a**), $H_{3}$ (**b**) and$\;H_{4}$ (**c**). A gray disk shows the deputy the carbon atomic layers nearest to the interface and a hollow circle shows the deputy the nitrogen atomic nearest to the interface. Red hollow circles express N atoms bonded to C and black hollow circles express N atoms with dan-gling bonds. Dotted wires represent cells in the flat surface. (**d**) Computed band alignment. The experimental band gaps ($E_{g}$) gave date of the average valence-band offset (VBO) and the corresponding conduction-band gaps (CBO) are shown.

**Figure 3 molecules-28-01334-f003:**
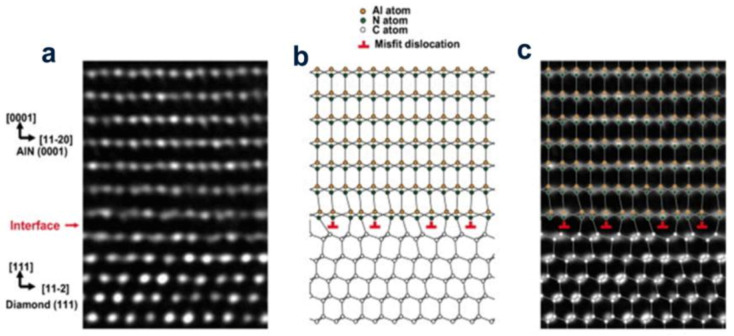
(Color online) (**a**) HR-TEM graphics of the AlN (0001)/diamond (111) heterogeneous contact surface. The small pointer displays the AlN/diamond contact surface. (**b**) Atomic distribution model obtained from the HR–TEM graphics. (**c**) is formed by (**a**,**b**).

**Figure 4 molecules-28-01334-f004:**
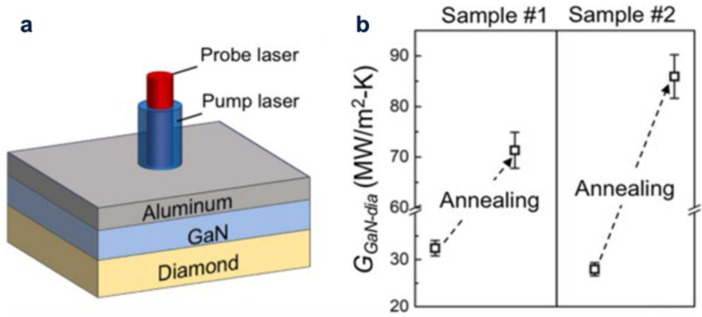
(**a**) The diagram of GaN/Al/diamond structure fathomed by TDTR, (**b**) The diagram thermal boundary conductance of diamond contact face (GGaN-dia)/GaN fathomed by TDTR.

**Figure 5 molecules-28-01334-f005:**
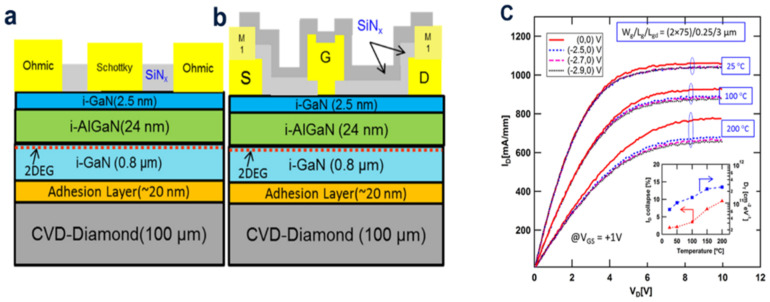
(**a**) Transverse-truncated surface sketch map of AlGaN/GaN: (**a**) Schottky diode, (**b**) The 0.25 μm palisade HEMTs on the CVD-diamond underlayment. (**c**) Temperature-dependent pulsed *I_DS_–V_DS_* peculiarity of HEMTs stressed under different static bias voltages. (Inset) Inner diameter collapse and DT variation at all sorts of temperatures (25–200 °C).

**Figure 6 molecules-28-01334-f006:**
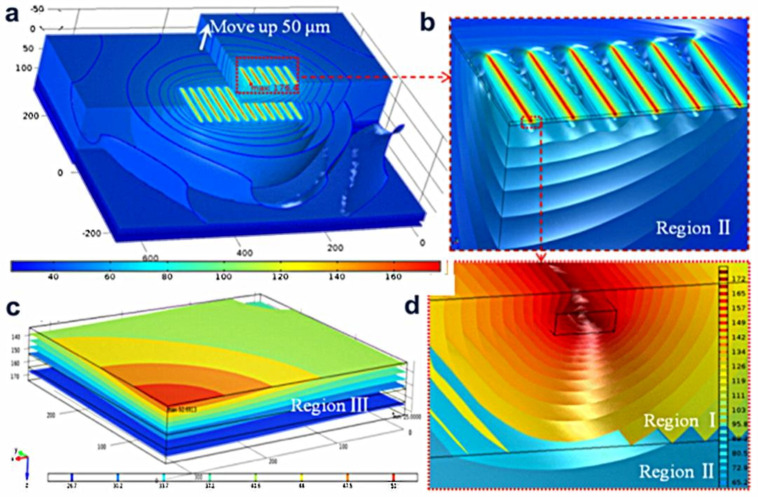
(**a**) The actual module, (**b**) Near-junction region, (**c**) Heat sources region, and (**d**) Temperature distribution and isothermal surface of solder and water tank.

**Figure 7 molecules-28-01334-f007:**
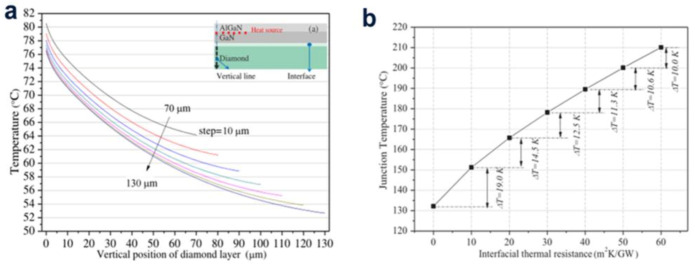
(**a**) Four different diamond thicknesses, (**a**) Sketch map of the perpendicular location, the temperature distribution in the perpendicular position of diamond substrate is simulated. (**b**) Relationship between thermal resistance of contact surface and junction temperature under some meaningful data.

**Figure 8 molecules-28-01334-f008:**
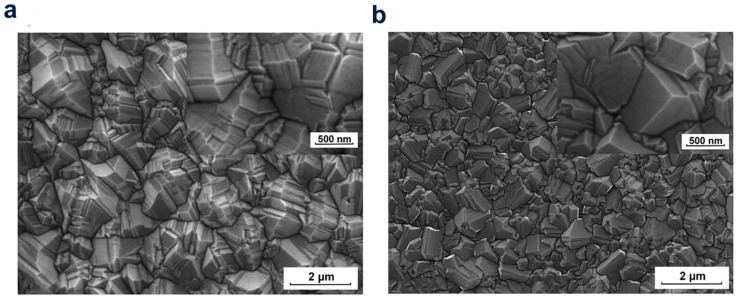
SEM micrographs of the diamond layers made of various craft gas ingredients: (**a**) DFI$CH_{3}$OH/$H_{2}$ = 1.0 vol.%, (**b**) DFII CH_3_OH/H_2_ = 4.0 vol%.

**Figure 9 molecules-28-01334-f009:**
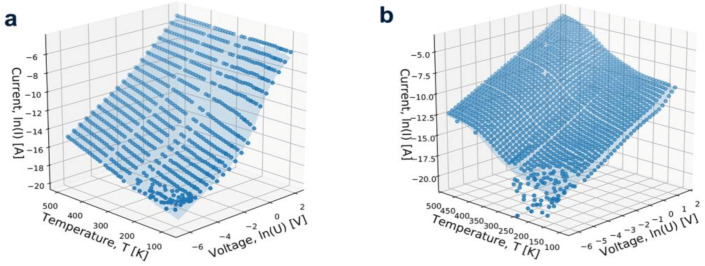
The *I-V-T* characteristics of the heterojunctions (**a**) DFI and (**b**) DFII collected in the forward configuration plot as a function of the voltage and the temperature indicate that the heterojunctions (**a**) DFI and (**b**) DFII have a supposedly calculated conducting face.

**Figure 10 molecules-28-01334-f010:**
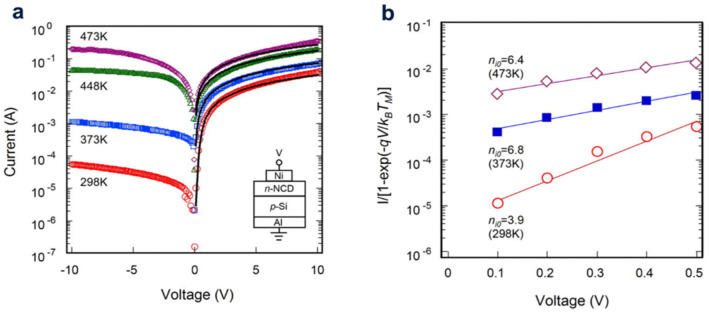
(**a**). At TM = 298, 373, 448, and 473 K, the series resistance RS used in the fitting were 36×, 130, 55, and 300, respectively. *I-V* measurement of the n-NCD/p–-Si heterojunction properties with TM variation. The real curve is obtained by using equation fitting at n_i_ = 2.4 for each TM. (**b**) Draw $1-\exp\left(-\frac{qV}{k_{B\;}T_{M}}\right)\;$curve of TM with V. When TM = 473, 373 and 298 K, the straight lines are fitting curves of n_i0_ = 6.4, 6.8 and 3.9, respectively.

**Figure 11 molecules-28-01334-f011:**
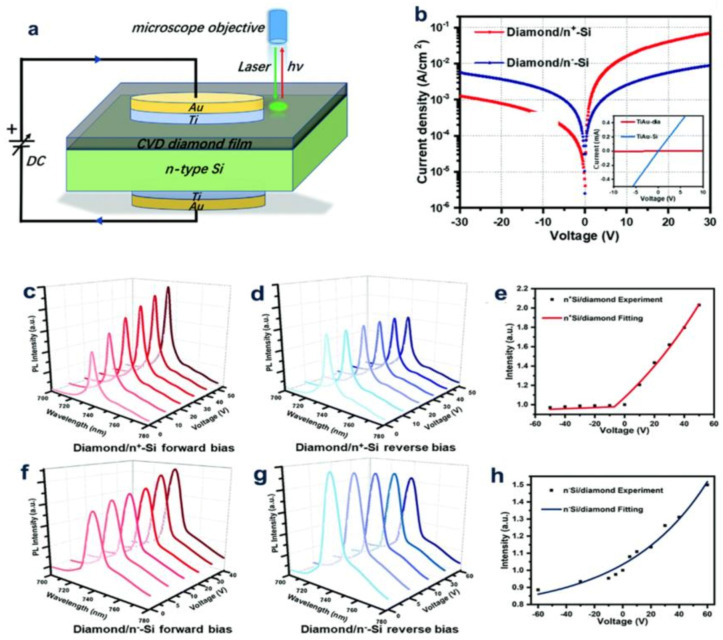
(**a**) Sketch map of making diamond-Si diode (**b**) corresponds to semilog *J–V* curves. The PL spectra of the SiV^−^ centers in the diamond/n+-Si (**c**–**e**) and diamond/n^-^-Si (**f**–**h**) heterojunctions under various bias voltages: (**c**,**f**) forward bias; and (**d**,**g**) reverse bias. (**e**,**h**) The plot of the SiV^-^ ZPL strength as a function of the bias voltage.

**Figure 12 molecules-28-01334-f012:**
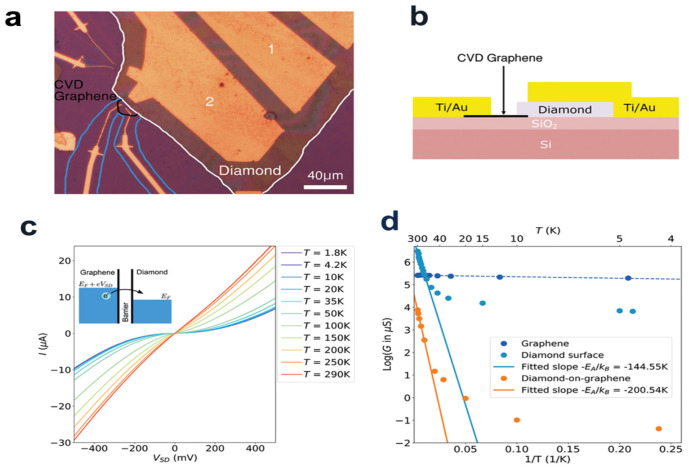
(**a**) Optical properties of a diamond-on-graphene heterojunction (evaporated contacts tagged via a blue line, the CVD graphene islands tagged via a black line, and the diamond nanosheet tagged via a white line). The data indicate the diamond-on-graphene heterojunction, diamond sample and graphene sample. (**b**) lateral view, sketch map of sample. Upper left illustrations: Tunneling behavior of heterojunction at low-temperature and low-bias voltage. (**c**) Temperature-dependent *I–V* trait of diamond heterostructures on graphene. The blue line is used to represent the cutting area of graphene island, which avoids cross current via etching a tiny zone. (**d**) Logarithm of conductance G versus *1/T* at the low bias *I–V* curves (to make it easy to read, including the top axis of T in K).

**Figure 13 molecules-28-01334-f013:**
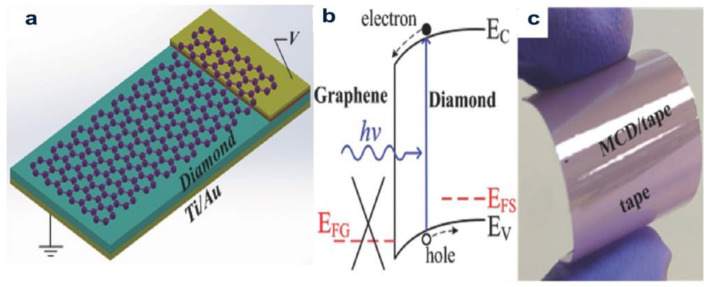
(**a**) Concept map of the solar-blind UV detector on the basis of the microcrystalline diamond-graphene heterojunction; (**b**) frequency band sketch map of the system under UV illumination; (**c**) Optical photograph of the tape/flexible MCD/metal bended between two human fingers [34].

**Table 1 molecules-28-01334-t001:** The related efforts of optoelectronic performance of non-metallic oxide/diamond pn heterostructures that need to be explored in depth.

Other Relevant Work Need to be Studied in Depth	
Performance Need to Raise:	High-Temperature Performance:
Seeking new manufacturing routes	Carrier mobility and generation rate
Seek new electrode types	Intrinsic carrier density
Raise the reaction speed	Current transmission mechanism
Combined with different device structures	Power loss and reducing circuit behavior

## Data Availability

The data presented in this study are contained within the article.

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
