# Peer review of "A Review on Optoelectronical Properties of Non-Metal Oxide/Diamond-Based p-n Heterojunction"

_molecules, 2023, doi:10.3390/molecules28031334_

Round 1
Reviewer 1 Report
The review manuscript entitled “A review on electronic and optoelectronic properties of non- metal oxide/diamond based pn heterojunction” is well-written, well-organized, and of publication quality. Different types of non-metal oxide/diamond-based heterojunction have been discussed and the study offers immense potential for futuristic diamond devices. Thus, it deserves publication in the “molecules” journal. I have a few minor comments as attached

Reviewer 2 Report
Comments and Suggestions for Authors
In this work entitled “A review on electronic and optoelectronic properties of non-metal oxide/diamond based pn heterojunction” by X Sang et al, the review is based on the progress in the device fabrication, optoelectronic performance research, LED, photodetector and HEMT of diamond-non metal oxide heterojunctions. The existing problems in the research of diamond/non-metallic oxide heterojunction optoelectronic devices are addressed, and the development potentials, which will promote the birth of new high-temperature and high-power optoelectronic devices are introduced. Accordingly, with the prime focus on the optoelectronic properties of the non-metal oxide/diamond, stability, carrier lifetime and their close correlation with the temperature factor are reviewed, and to overcome the power loss issue reasonable solutions are suggested.
Therefore, I would recommend publication pending minor revisions:
1. To clearly identify the paper's aims, modify the title to “A review on optoelectrical properties of non-metal oxide/diamond-based p-n heterojunction”
2. Line 41, please explain whether the "small gap of diamond" refers to the “small grain boundaries of diamond”, or “stronger atomic bounds”, etc. Also in Line 127, if the word "of set" is meant to be "offset," please edit.
3. There are still publications in recent years that have not been included in this review; given that the major purpose is to evaluate the recent ten years of research breakthroughs, please include a table highlighting the other relevant works that have not been studied in depth here.
Thanks.
